# Virulent Phage vB_EfaS_WH1 Removes *Enterococcus faecalis* Biofilm and Inhibits Its Growth on the Surface of Chicken Meat

**DOI:** 10.3390/v15051208

**Published:** 2023-05-20

**Authors:** Xinxin Jin, Xiuxiu Sun, Zui Wang, Junfeng Dou, Zhengdan Lin, Qin Lu, Tengfei Zhang, Guoyuan Wen, Huabin Shao, Guofu Cheng, Qingping Luo

**Affiliations:** 1Key Laboratory of Prevention and Control Agents for Animal Bacteriosis (Ministry of Agriculture and Rural Affairs), Institute of Animal Husbandry and Veterinary, Hubei Academy of Agricultural Sciences, Wuhan 430064, China; jinxinxin@webmail.hzau.edu.cn (X.J.); wangzui@webmail.hzau.edu.cn (Z.W.); djf0825@163.com (J.D.); luqin@webmail.hzau.edu.cn (Q.L.); tfzhang23@163.com (T.Z.); wgy_524@163.com (G.W.); shhb1961@163.com (H.S.); 2College of Veterinary Medicine, Huazhong Agricultural University, Wuhan 430070, China; xiu807313@webmail.hzau.edu.cn (X.S.); lzd0613@sina.com (Z.L.); 3Hubei Provincial Key Laboratory of Animal Pathogenic Microbiology, Institute of Animal Husbandry and Veterinary, Hubei Academy of Agricultural Sciences, Wuhan 430064, China

**Keywords:** *Enterococcus faecalis*, bacteriophage, biofilm, genome, biocontrol agent

## Abstract

*Enterococcus faecalis* is a potential animal and human pathogen. Improper use of antibiotics encourages resistance. Bacteriophages and their derivatives are promising for treating drug-resistant bacterial infections. In this study, phylogenetic and electron microscopy analyses of phage vB_EfaS_WH1 (WH1) isolated from chicken feces revealed it to be a novel phage in the family *Siphoviridae*. WH1 showed good pH stability (4–11), temperature tolerance (4–60 °C), and broad *E. faecalis* host range (60% of isolates). Genome sequencing revealed a 56,357 bp double-stranded DNA genome with a G+C content of 39.21%. WH1 effectively destroyed *E. faecalis* EF01 biofilms, even at low concentrations. When WH1 was applied at 1 × 10^5^ to 1 × 10^9^ PFU/g to chicken breast samples stored at 4 °C, surface growing *E. faecalis* were appreciably eradicated after 24 h. The phage WH1 showed good antibacterial activity, which could be used as a potential biocontrol agent to reduce the formation of *E. faecalis* biofilm, and could also be used as an alternative for the control of *E. faecalis* in chicken products.

## 1. Introduction

*Enterococcus faecalis* is a major bacterium in the human and animal intestines. However, it is also considered an opportunistic pathogen [1]. *E. faecalis* often causes septicemia, root canal infection, and endocarditis, which endanger human and animal health [2,3,4].

*E. faecalis* has a strong adaptability and resistance to the environment, so it is difficult to be effectively treated in a clinical setting [5,6]. Enterococcus strains have been isolated from chicken and pork sold in retail locations in Sweden and Denmark [7,8]. Bacterial biofilm refers to a large number of bacterial aggregation membrane-like substances formed by bacteria adhering to the contact surface, secreting polysaccharide matrix and fibrin and lipid proteins, and wrapping themselves around it. Bacteria in biofilms are protected from the influence of antibiotics, which leads to infection that is difficult to treat [9]. Therefore, the elimination of *E. faecalis* biofilm has a positive effect on clinical treatment.

Bacteriophages are a class of viruses that attack bacteria and are highly specific [5]. Only a few studies have examined the effectiveness of phages and their derivatives in controlling biofilms formed by *E. faecalis* [10,11]. There is evidence that the drug resistance rate of *E. faecalis* in broiler chicken farms is high and there is a risk of vertical transmission [12].

In this study, we isolated bacteriophage vB_EfaS_WH1 (hereinafter referred to as WH1) from chicken feces using *E. faecalis* strain EF01 as the host bacterium. Electron microscopy, biological characterization, and whole-genome sequence analyses of this novel phage were performed. The findings implicate WH1 as a possible approach to remove *E. faecalis* biofilm and control *E. faecalis* contamination in chicken meat.

## 2. Materials and Methods

### 2.1. Bacterial Strains and Media

*E. faecalis* strain EF01 isolated from chicken livers was stored in the Pathology Laboratory of Huazhong Agricultural University, Wuhan, China. We tested the bacteriophage host range by spot assay using a laboratory collection of 30 bacterial strains, including *Staphylococcus*, *Escherichia coli*, *Salmonella*, and *Enterococcus*. The strains were cultured overnight in tryptic soy agar (TSA, BD Difco, NJ, USA) or in tryptic soy broth (TSB, BD Difco) at 37 °C

### 2.2. Isolation and Purification of WH1

Bacteriophage WH1 was isolated from the intestinal contents of chickens. Chicken manure was resuspended in sterile SM buffer (100 mM NaCl, 8.5 mM MgSO_4_·7H_2_O, and 50 mM Tris-Cl (pH 7.5)) overnight at 4 °C, as described previously [13]. The SM solution was centrifuged for 10 min at 10,000× *g*. The collected supernatant was filtered through a 0.22 μm Millex–GP microporous membrane (Millipore, MA, USA) to remove bacteria. A total of 200 μL of the filtrate and 200 μL of exponentially growing *E. faecalis* EF01 were mixed in 5 mL TSB medium and incubated for 12 h at 37 °C [14]. The culture was centrifuged for 2 min at 12,000× *g* to remove the bacteria. The supernatant was filtered through another 0.22 μm Millex–GP microporous membrane. Subsequently, three rounds of phage purification were performed using the double-layer ager method [15]. The resulting phage stock was stored at 4 °C until used for further experiments.

### 2.3. Determination of Host Range of WH1

The host range of the purified WH1 against the 30 bacterial indicator strains included 20 strains of *Enterococcus faecalis*, 1 strain of *Enterococcus faecium*, 1 strain of *Clostridium perfringens*, 1 strain of *Staphylococcus aureus*, 2 strains of *Escherichia coli*, and 4 strains of *Salmonella*, and 1 strain of *Klebsiella* (Table 1) was determined by the spot method [16]. A total of 100 μL of each newly cultured indicator strain was mixed with 5 mL of 0.5% TSA soft agar at 50 °C and then poured on the surface of a plate of pre-prepared 1% TSA agar. Ten microliters of purified phage WH1 (10^8^ plaque-forming units, PFU) was placed on a double-layer plate and the plates were incubated for 14 h at 37 °C. The susceptibility of bacteria to WH1 infection was confirmed by the presence of clear plaques. The results were divided into plaque-positive (+) and plaque-negative (−) groups.

### 2.4. Biological Characteristics of WH1

The pH of TSB liquid medium was adjusted to a range of 2.0 to 12.0. A total of 100 μL phage (1 × 10^8^ PFU/mL) was mixed with 900 μL TSB. The phage titer was determined after 1 h of incubation at 37 °C by the double-layer agar (DLA) method [17].

To explore the temperature sensitivity of WH1, 100 μL of the phage suspension containing (1.6 × 10^8^ PFU/mL) was mixed with 900 μL TSB and incubated at 70, 60, 50, 37, 28, and 4 °C for 20, 40, 60, and 80 min for each temperature. The phage titer was determined by the DLA method.

According to the specific multiplicity of infection (MOI, 0.0001, 0.001, 0.01, 0.1, 1, 10, and 100), purified WH1 and the host bacteria were mixed together and cultured for 6 h at 37 °C with shaking at 220 rpm. The mixture was centrifuged for 5 min at 12,000× *g*. The precipitate was discarded and the supernatant was filtered through a 0.22 μm filtrate [18]. The phage titer of different MOIs from samples was determined by the DLA method. Each experiment had three biological repetitions.

Chloroform was mixed with WH1 in the ratio of 0%, 10%, 20%, 30%, 40%, and 50%. Each mixture was incubated for 1 h at 37 °C with shaking at 220 rpm. The phage titer was determined by a DLA method [19]. Each experiment was repeated three times.

### 2.5. One-Step Growth Curve of WH1

The one-step growth curve protocol has been described previously [20]. Briefly, *E. faecalis* EF01 was cultured to the logarithmic stage (optical density at 600 nm (OD600) = 0.5). The bacteria were collected by centrifugation for 15 min at 10,000× *g* and 4 °C and then suspended in fresh TSB. The bacteria were infected with purified WH1 (MOI of 0.01) and incubated for 5 min at 37 °C. After centrifugation, the supernatant containing unbound phage was discarded. The pellet was suspended in 10 mL of fresh TSB and oscillated for 120 min at 37 °C and 220 rpm. Samples were collected every 10 min and the phage titer was determined by the DLA method. Each experiment had three biological repetitions. The phage burst size was calculated as the ratio of the average phage titer value of the plateau to the average titer value of the incubation period [21].

### 2.6. Phage DNA Extraction and Genome Sequence Analysis

Genomic DNA of phage WH1 was extracted by an established phenol–chloroform method [17,22]. The Illumina NovaSeq platform 2500 (Shanghai Majorbio Bio-Pharm Technology Co., Ltd. The equipment was purchased from illumina, San Diego, CA, USA) was successfully used to sequence the genomic DNA of phages. Bcl2fastq (v2.17.1.14) software was used for preliminary quality analysis and raw sequencing data were obtained, as previously described [23]. GeneMarkS software (http://topaz.gatech.edu/GeneMark/ accessed on 5 February 2023) was used to predict the protein coding genes of the genome and open reading frames (ORFs) in WH1 [24]. Diamond software was used to compare the coding genes [25]. Bacteriophage resistance and virulence factors in ORFs were analyzed using an antibiotic resistance gene database (https://card.mcmaster.ca/ accessed on 5 February 2023) and Virulence Factors of the Pathogenic Bacteria database (http://www.mgc.ac.cn/cgi-bin/VFs/v5/main.cgi accessed on 5 February 2023). PHASTER was used to predict integrase-related genes and their attachment sites [26].

The CGView server was used to create circular map of WH1 genome [27,28]. The phylogenetic tree of the large subunit amino acid sequence of phage terminase was constructed using the neighbor-joining method in MEGA 7 [29]. This tree was drawn to scale, and the unit used to infer the evolutionary distance of phylogenetic tree was the same as the branch length. The evolutionary distance was calculated by the P-distance method [30]; the units are the number of differences per site base. Easyfig 2.2.5 (http://mjsull.github.io/Easyfig/ accessed on 15 February 2023) was used to map the genome-wide collinearity ratio [31].

### 2.7. Transmission Electron Microscopy TEM of WH1

The morphology of purified WH1 phage was observed by TEM. Twenty-five microliters of a suspension of purified WH1 was placed on a carbon-coated copper grid for 15 min. Phage was negatively stained with 2% (*w*/*v*) phosphotungstic acid for 3 min and observed with a transmission electron microscope (JEOL, Tokyo, Japan) at 80 kV, as previously described [1]. The phage was classified according to the guidelines of the International Committee on Virus Taxonomy (ICTV) [32].

### 2.8. Effect of WH1 Phage on Biofilms of E. faecalis

*E. faecalis* EF01 biofilms were prepared as previously described [33,34]. Strain EF01 was inoculated in 5 mL TSB and cultured overnight at 37 °C. The culture was diluted to OD600 of 0.2. Two milliliter aliquots of the dilute bacterial suspension were added to wells of a 12-well plate and cultured for 24 h at 37 °C to permit formation of biofilms. The supernatant was removed from each well and the biofilm was washed four times with 1 mL of sterile phosphate-buffered saline (PBS) to remove floating bacteria. WH1 was then added to each well except the control group to achieve a final titer of 10^2^, 10^4^, 10^6^, or 10^8^ PFU per well and incubated for 24 h at 37 °C. Phage and bacteria suspensions were discarded and the wells were washed four times with 1 mL of PBS each time. The plate was air-dried for 10 min at 37 °C and the wells were stained with 1 mL of 0.1% crystal violet (CV) at room temperature for 25 min. Each biofilm was washed four times with 1mL volumes of PBS to remove the excess CV. One milliliter of 95% ethanol was added to each well, and the plate was placed on a shaker at room temperature for 15 min to dissolve the CV bound to the biofilm. The optical density at 600 nm was measured using a Multiskan FC table reader (Thermo Fisher Scientific, Waltham, MA, USA). We also measured the amount of bacteria in different groups. This experiment was repeated three times.

### 2.9. Inactivation of E. faecalis in Chicken Breast by Phage WH1

To minimize contamination by spoilage organisms, raw chicken breasts were aseptically cut into slices on a sterile lab bench. Each chicken breast was aseptically cut into 2 cm × 2 cm squares (about 1 g), and sterilized twice with ultraviolet light, as previously described [35,36]. Before artificial inoculation, all samples were inoculated on TSA to ensure there were no microorganisms [37]. In the experiment, the sterile chicken pieces were soaked in a culture of *E. faecalis* EF01 (approximately 1 × 10^5^ colony forming units (CFU)/g) for 15 min [38]. The breast tissues were placed in wells of a 24-well plate. The surfaces of these tissues were then treated with phage WH1 by dispensing 0.2 mL of a phage suspension containing 10^5^, 10^7^, or 10^9^ PFU/g onto the tissues at 4 °C.

An unsoaked tissue square in the bacterial culture was used as the phage control. Tissues that were not treated with WH1 were used as the bacterial control group. The 24-well plates were immediately sealed in polystyrene Petri dishes and incubated for 1 day at 4 °C. At 0, 1, 6, 12, and 24 h, *E. faecalis* content in chicken tissue sample was determined as CFU. Each sample was prepared in triplicate.

### 2.10. Statistical Analysis

One-way analysis of variance (ANOVA) was used to evaluate the difference between the experimental and control groups. All experiments were repeated three times with similar results. GraphPad Prism 7.0 (GraphPad Software, San Diego, CA, USA) was used to plot the data. Significance was evident at *p* < 0.05, *p* < 0.01, or *p* < 0.001, depending on the experiment.

## 3. Results

### 3.1. Isolation and Identification of E. faecalis Phage

*E. faecalis* EF01 was used as the host to form transparent and uniform plaques on a DLA plates. The phages were isolated by purifying the plaques four times. TEM determined that WH1 phage particles had an oblate head (length 120 nm ± 2; width 5 nm ± 1) and long tail (180 mm ± 2) (Figure 1A). According to ICTV guidelines, the isolated phage belongs to the *Siphoviridae* family of viruses. It was designated vB_EFaS_WH1 (WH1).

### 3.2. Host Range of Phage WH1

A host range analysis revealed that WH1 could form transparent phagocytic circles when incubated with the 12 selected strains of *E. faecalis*. The cleavage rate was 60% (12/20). However, WH1 did not lyse bacteria from other genera (Table 1). WH1 did not infect all of the tested Gram-negative bacteria (*E. coli*, *Klebsiella pneumoniae*, and *Salmonella*) and three of the Gram-positive strains (*Enterococcus faecium*, *Clostridium perfringens*, and *Staphylococcus aureus*). The findings were evidence of the wide host range of phage WH1.

### 3.3. Biological Characteristics of the Phage WH1

To evaluate heat resistance, WH1 was incubated at 4, 28, 37, 50, 60, and 70 °C for up to 80 min. The titer of phage WH1 was stable between 4 and 50 °C, it decreased significantly at 60 °C, and the phage was rapidly deactivated at 70 °C (Figure 1B).

The influence of different pH values on WH1 infectivity was assessed. The results of the acid–base tolerance test showed that WH1 was stable at pH 4.0–11.0. The highest titer obtained occurred at pH 7.0 (Figure 1C). When the MOI was 0.1, the highest number of phage WH1 was 2.0 × 10^9^ PFU/mL. Thus, this MOI was optimal for phage WH1 (Figure 1D).

The results of the phage chloroform sensitivity test are shown in Figure 1F. When <30% chloroform was added, the phage titer was higher. When >40% chloroform was added, the titer of the phage decreased significantly, but it did not lead to inactivation.

### 3.4. One-Step Growth

The one-step growth curve of WH1 on *E. faecalis* EF01 was detected to determine the latent period and burst size of WH1 (Figure 1E). The latent period of phage WH1 was 10 min, followed by a burst period of 40 min. WH1 plateaued after 50 min. The average burst size was approximately 70 PFU per cell.

### 3.5. Whole-Genome Sequencing and Coding Gene Prediction of Phage WH1 Genome

Whole-genome sequencing analysis of WH1 revealed a double-stranded DNA genome with a length of 56,357 bp and a G+C content of 39.21% (Figure 2). BLAST analysis of nucleic acid revealed 95.81% homology of WH1 with the published *E. faecalis* phage EF-P29 (GenBank: KY303907.1) and 95.69% homology with EF-P10 (GenBank: KY472224.1).

Phage WH1 had 90 ORFs. Of these, 31 had established homology with the functional proteins annotated in the NCBI database, and the remainder were homologous with a hypothetical protein (Table 2). There were eight putative proteins involved in the process of phage DNA packaging and replication: terminase small subunit (ORF 40), terminase large subunit (ORF 43), portal protein (ORF 45), transfer RNA (tRNA; Trp-CCA, ORF 67), DNA primase (ORF 71), DNA replication protein (ORF 73), replicative DNA helicase (ORF 74), and DNA polymerase (ORF 88). The phage genome invasion process is mediated by a DNA channel called a portal protein (ORF 45), through which phage DNA is injected into the host and is involved in protein connections between the phage head and tail. ORF 60 was 99.16% homologous to the published *E. faecalis* phage vB_EfaS_IME198 lysin (GenBank: YP_009218898.1).

In those structural proteins, ORF48 encodes the phage major head protein. ORF49 encodes the main tail protein of the phage, whereas ORF57 encodes the tail fiber of the phage. The binding of bacteriophages to the host bacteria was mediated by these structural proteins. Among the lysis proteins, ORF42 is interpreted as holin, which can form tiny pores in the inner membrane. ORF60 is associated with lysis of the peptidoglycan layer of the host cell wall. ORF76 encodes HNH homing endonuclease, an intron protein with sequence tolerance and site specificity. Glutaredoxin (ORF 35) and adenylate kinase (ORF 80) were also annotated in WH1. No genes encoding virulence factors or antibiotic resistance and genes related to lysogenicity were predicted, suggesting that WH1 can be used for the treatment of *E. faecalis*-related diseases. The phylogenetic tree of WH1 was constructed based on the nucleotide sequence of the terminase large subunit (ORF 43). Multiple genome alignments and phylogenetic tree analyses of phage WH1 and *E. faecalis* EFKL, IME-EF1, EF653P3, EF653P1, PHB08, SSsP-1, EF1c55, Ef2.2, EF653P5, IME198, and EFC1 are shown in Figure 3. The results showed that the whole genome of phage WH1 was highly homologous with the phage IME198 genome. Therefore, IME198 and WH1 were selected for collinearity analysis. Phages WH1 and IME198 showed extremely high collinearity (Figure 4). We found that some genes were located on opposite strands of the DNA, possibly due to genetic rearrangement that allowed the phage to better adapt to its environment.

No homologs of phage transposases, repressors, integrases, and excision enzymes were predicted in the WH1 genome. Based on the sequencing results, phage WH1 can be considered a new phage.

### 3.6. Effect of Phage WH1 on E. faecalis Biofilm

To detect whether phage WH1 can destroy the biofilm formed by *E. faecalis* EF01, the CV method was used to stain the biofilms after phage treatment. Phage WH1 significantly reduced biofilm formation, even at very low numbers of 10^2^ plaque-forming units (Figure 5A,B). The finding provided evidence that phage WH1 can effectively reduce the biofilm formed by *E. faecalis* EF01. The number of bacteria in the phage addition group was significantly reduced compared to the control group (Figure 5C).

### 3.7. Inactivation of E. faecalis in Chicken Breast Using Phage WH1

Phage WH1 showed a typical dose-dependent inhibition of *E. faecalis* EF01 (Figure 6). The concentration of *E. faecalis* decreased by approximately 2.7 log after adding 10^9^ PFU bacteriophage for 24 h. As the phage dose decreased, the inhibition efficiency decreased. The concentration of *E. faecalis* in the control samples without phage treatment approached the initial concentration after 24 h of incubation.

The decrease in *E. faecalis* counts were directly proportional to the phage density; the decrease in the *E. faecalis* count was greater with the increase in phage concentration.

## 4. Discussion

Phages are the most common and widely distributed group of viruses. Phages have been used in different industries and fields, such as medicine, environment, and livestock production, as well as in aquaculture and food processing [39].

Bacteriophage therapy for *E. faecalis* infectious diseases has achieved good results [40,41]. In this study, phage WH1, isolated by us, was stable at temperatures ranging from 4 to 60 °C and pHs between 4.0 and 11.0. This stability will be convenient for storage and transportation for future clinical applications. One-step growth experiments showed an average burst size of 70 PFU per cell, which is consistent with the average number of commonly reported phage bursts (approximately 30 PFU–120 PFU/cell per cell) [42]. The morphology of phage WH1 revealed by TEM was typical of the *Siphoviridae* phage. Most of the *E. faecalis* phages that have been previously isolated belong to the *Siphoviridae* family [43,44]. The phage WH1 was sensitive to chloroform, and we speculated that the capsid might contain liposomes.

Genomics is one of the effective methods for understanding the characteristics of phages at the molecular level. The genetic diversity and similarity of genomes are closely related to the lytic ability of phages [45]. The terminase large subunit gene of phages is relatively conserved. The tail protein affected the tail structure of phages and also played an important role in bacteriophage infection [46,47]. The tail fiber of phages contains receptor-binding proteins that bind with cell surface receptors [48]. No presumed antibiotic resistance genes or virulence factors were found by a genome sequencing analysis. This is an interesting discovery. Phage WH1 has a broad cleavage spectrum, which is related to the fact that its tail fiber can recognize a variety of host receptor proteins and may also be particularly effective in the lysin encoded by ORF60 in the WH1 genome. Holin (ORF42) is a perforin protein that penetrates the cell membrane and inserts itself into bacteria [49]. Therefore, WH1 may use the holin–endolysin tactics to lyse the host cell and release lytic virions [50].

Most bacterial infections are related to biofilms, which can hamper treatment [51,52]. Phages and their lyases can degrade biofilms [15]. In this study, phage WH1 displayed an excellent ability to destroy *E. faecalis* biofilm, which is an equally efficient ability to that of phage EFDG1 [53]. Our results showed that biofilms were inhibited regardless of the phage titer. Phages can reduce the number of bacteria, thus affecting the formation of biofilm, indicating that a phage has the ability to inhibit the formation of biofilms. In the future, we will further investigate the mechanism of action of phages and biofilm exopolysaccharides.

Concerning bacterial resistance, the appropriate phage must be isolated. A “cocktail therapy” involving a phage is a potential candidate for treating infections caused by antibiotic-resistant bacteria [54,55]. The stability of phages is an important parameter for the manufacture of phage preparation [56]. We observed that WH1 is relatively stable at temperatures ranging from 4 to 60 °C, which could permit the manufacturing of the phage. Another study demonstrated that phage cocktails reduced *Salmonella* contaminations in chicken farms [57]. Previous studies have shown that phages ingested by healthy individuals are safe, active, and they accumulate in the lower intestine, inhibiting human IBD-associated gut microbes [58]. A review of phage therapy shows that phage therapy is generally safe in 20 animal studies, 35 clinical case reports, and 14 clinical trials [59]. The collective findings indicate the future suitability of phage WH1 for reducing pathogenic *E. faecalis* contamination in farm locations, such as chicken coops or egg incubators.

The present study also demonstrates WH1-mediated inactivation of *E. faecalis* in chicken breast. With increasing WH1 concentrations, the number of viable *E. faecalis* decreased significantly. This result is similar to the results of phage inhibition of *Listeria monocytogenes* in fish and phage inhibition of *Clostridium perfringens* in chicken [35,57]. Our results demonstrated that phage WH1 can inhibit bacteria at 4 °C, and it is speculated that phages can still recognize and adsorb sensitive bacteria at 4 °C, suggesting that phages can control pathogenic bacteria in frozen meat products.

We evaluated the phage WH1 as a potential biocontrol agent for reducing the biofilm formation of *E. faecalis* and as an alternative for the prevention and control of *E. faecalis* in the chicken.

## 5. Conclusions

In summary, we isolated a virulent phage (WH1) that targets *E. faecalis*. The findings indicate the potential value of WH1 as a biocontrol agent for reducing biofilm formation of *E. faecalis* and as an alternative for the prevention and control of *E. faecalis* contamination of chicken meat. Our future work will address the treatment of bacterial infections with WH1 lyase and the replacement of antibiotics.

## Figures and Tables

**Figure 1 viruses-15-01208-f001:**
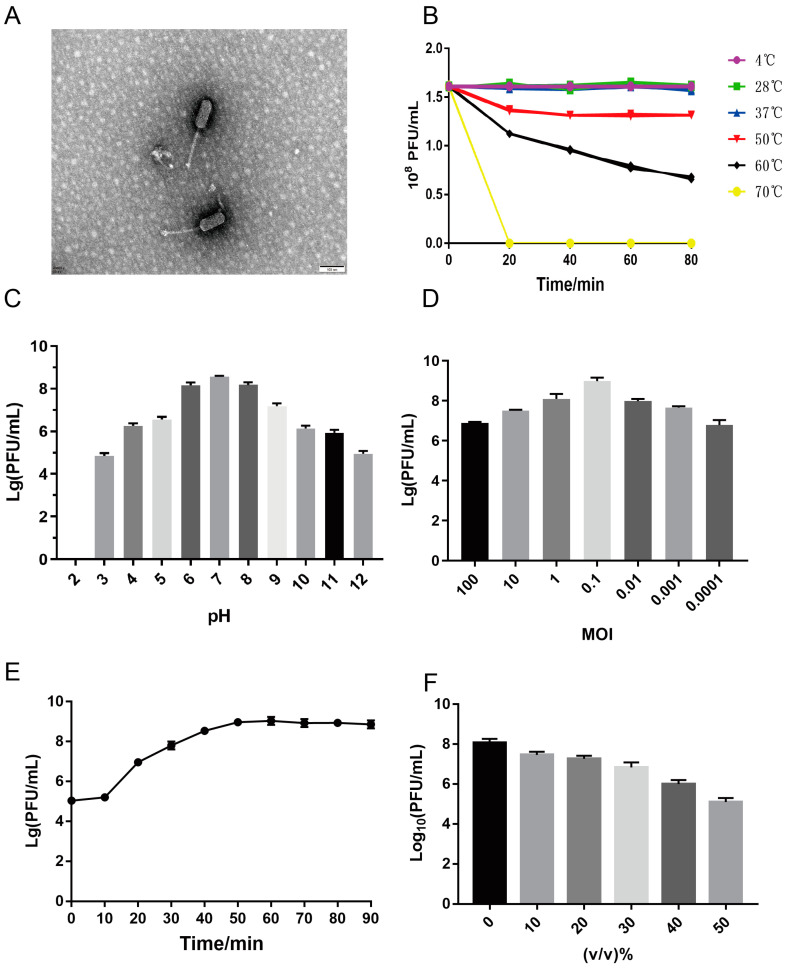
Biological characteristics of phage WH1. (**A**) Transmission electron microscopy observation of phage vB_EfaS_WH1. The scale bar is 100 nm. (**B**) Temperature stability of phage WH1. (**C**) Stability of phage WH1 at different pHs. (**D**) Optimal multiplicity of infection (MOI) of phage WH1. (**E**) One-step growth curve of phage WH1 in host strain *E. faecalis* EF01. (**F**) Effect of chloroform on phage WH1.

**Figure 2 viruses-15-01208-f002:**
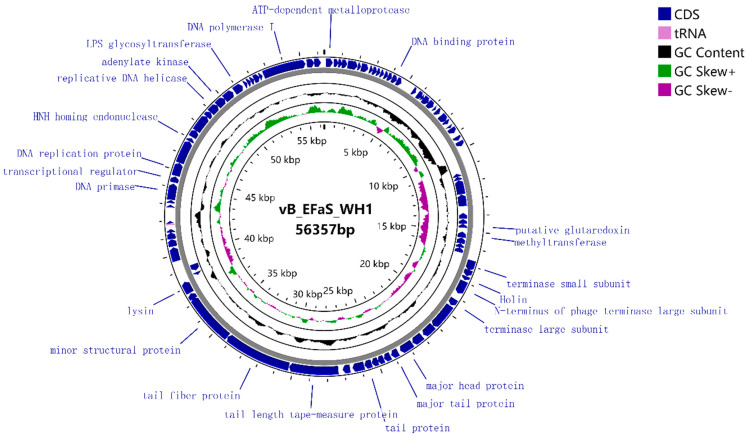
Genome map of phage WH1 generated by CGView. The functions of open reading frames are annotated using BLASTP and the Rast database. The regions in blue represent the distribution of the coding sequence (CDS) region and the arrows indicate the direction of transcription. The total GC content (39.21%) is indicated in black, while the inner ring with green and purple histograms indicates GCskew. For clarity, the hypothetical protein is not described on the map.

**Figure 3 viruses-15-01208-f003:**
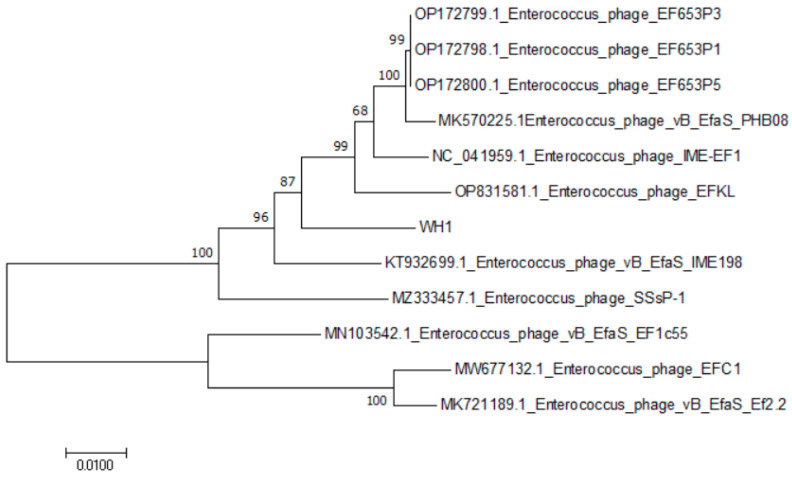
Cluster analysis of *E. faecalis* phage WH1. The phylogenetic tree was constructed using the neighbor-joining method based on the terminase large subunit.

**Figure 4 viruses-15-01208-f004:**
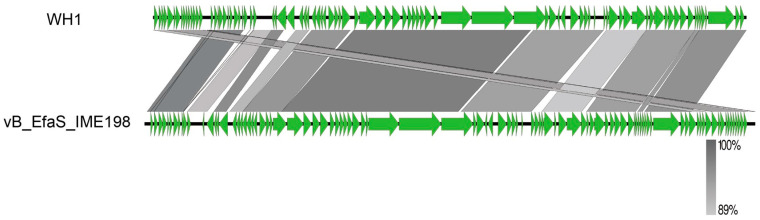
Genome homology analysis of phage WH1 with vB_EfaS_IME198. The green arrows indicates the open reading frames. The color gradient indicates the level of nucleotide identity between the phage genomes.

**Figure 5 viruses-15-01208-f005:**
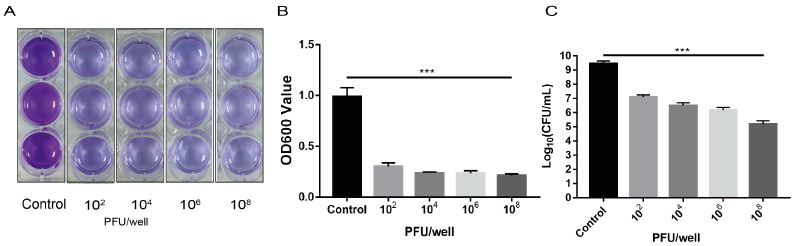
The initial titer of phageWH1 was 10^2^, 10^4^, 10^6^, or 10^8^ PFU/well. The biofilm biomass was obtained after incubation for 24 h. (**A**) Crystal violet staining analysis. (**B**) Optical density values measured at 600 nm. (**C**) The result of the plate count. The results are expressed as the mean ± SD (standard deviation) of three independent experiments. The t-test method was used to detect significant differences between control and test samples. *** *p* < 0.001.

**Figure 6 viruses-15-01208-f006:**
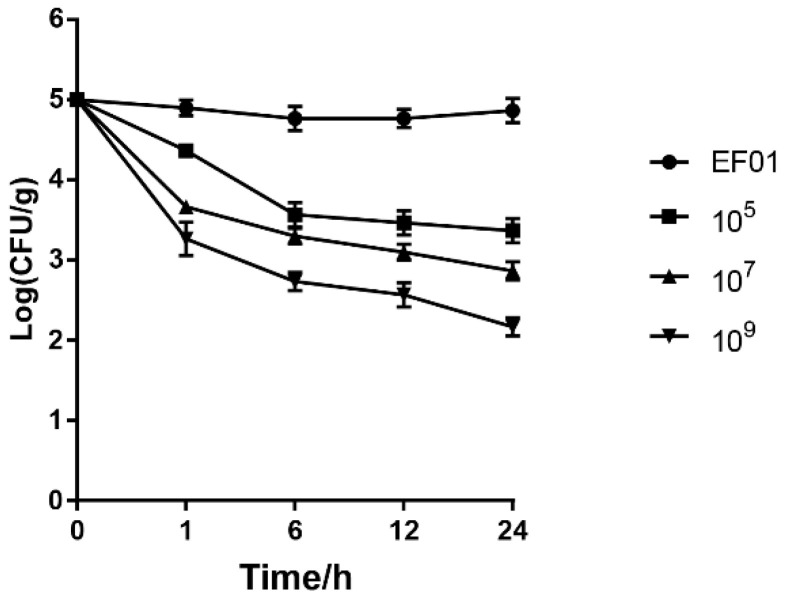
Inactivation of *E. faecalis* EF01 in chicken meat by WH1 at 4 °C. The data are the average of three separate experiments.

**Table 1 viruses-15-01208-t001:** Lytic activity of WH1 against the tested strains of *Enterococcus faecalis*.

NO.	Strains of Bacteria	Phage Sensitivity ^a^	Source
1	*Enterococcus faecalis* E1	+	Clinical isolate
2	*Enterococcus faecalis* A1	+	Clinical isolate
3	*Enterococcus faecalis* A2	−	Clinical isolate
4	*Enterococcus faecalis* A4	−	Clinical isolate
5	*Enterococcus faecalis* A5	+	Clinical isolate
6	*Enterococcus faecalis* A7	−	Clinical isolate
7	*Enterococcus faecalis* A8	+	Clinical isolate
8	*Enterococcus faecalis* A9	−	Clinical isolate
9	*Enterococcus faecalis* A10	+	Clinical isolate
10	*Enterococcus faecalis* A11	+	Clinical isolate
11	*Enterococcus faecalis* A12	+	Clinical isolate
12	*Enterococcus faecalis* A13	+	Clinical isolate
13	*Enterococcus faecalis* A14	+	Clinical isolate
14	*Enterococcus faecalis* A15	+	Clinical isolate
15	*Enterococcus faecalis* A16	−	Clinical isolate
16	*Enterococcus faecalis* N16	−	Clinical isolate
17	*Enterococcus faecalis* N17	−	Clinical isolate
18	*Enterococcus faecalis* N18	+	Clinical isolate
19	*Enterococcus faecalis* N19	−	Clinical isolate
20	*Enterococcus faecalis* N20	+	Clinical isolate
21	*Enterococcus faecium* Z3	−	Clinical isolate
22	*Clostridium perfringens*	−	ATCC13124
23	*Staphylococcus aureus*	−	ATCC25923
24	*Escherichia coli*	−	ATCC25922
25	*Escherichia coli* O157:H7	−	ATCC
26	*Salmonella* enteritidis	−	CVCC3375
27	*Salmonella* pullorum	−	CVCC529
28	*Salmonella* pullorum	−	CVCC530
29	*Salmonella* typhimurium	−	ATCC14028
30	*Klebsiella pneumoniae* Y1	−	Clinical isolate

^a^ Symbols: After the bacteria to be tested are infected with vB_EfaS_WH1 phage, the (+) region is clear zones or (−) has no plaques.

**Table 2 viruses-15-01208-t002:** Analysis of the main proteins of vB_EfaS_WH1.

ORF	Start	Stop	Strand	Function
ORF01	107	499	+	ATP-dependent metalloprotease
ORF05	1372	1956	+	Cytidine deaminase
ORF14	4085	4390	+	DNA-binding protein
ORF33	13,854	14,219	−	ABC transporter
ORF35	14,543	14,803	−	Putative glutaredoxin
ORF36	15,441	15,827	−	Methyltransferase
ORF40	16,679	17,278	+	Phage terminase small subunit
ORF42	17,655	17,903	+	Holin
ORF43	17,966	18,769	+	N-terminus of phage terminase large subunit
ORF44	19,144	19,560	+	Terminase large subunit
ORF45	19,617	21,152	+	Portal protein
ORF46	21,164	21,919	+	Head morphogenesis protein
ORF47	22,030	22,698	+	Head scaffolding protein
ORF48	22,747	23,553	+	Major head protein
ORF49	23,708	24,145	+	Major tail protein
ORF50	24,208	24,612	+	Head–tail connector family protein
ORF53	25,383	25,817	+	Phage tail protein
ORF54	25,838	26,530	+	Major tail protein
ORF56	27,359	30,244	+	Tail length tape-measure protein
ORF57	30,258	34,250	+	Tail fiber protein
ORF58	34,263	37,286	+	Phage minor structural protein
ORF60	37,635	38,351	+	Lysin
ORF71	43,256	44,200	+	DNA primase
ORF72	44,275	44,628	+	Transcriptional regulator
ORF73	44,677	45,453	+	DNA replication protein
ORF74	45,465	46,829	+	Replicative DNA helicase
ORF76	47,138	47,578	+	HNH homing endonuclease
ORF79	48,931	49,500	+	Crossover junction endodeoxyribonuclease
ORF80	49,497	50,066	+	Adenylate kinase
ORF82	50,783	51,346	+	LPS glycosyltransferase
ORF88	52,734	55,265	+	DNA polymerase I

## Data Availability

The dataset presented in this study can be accessed in an NCBI online repository (accession number: PRJNA939409).

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
