# Peer review of "Virulent Phage vB_EfaS_WH1 Removes Enterococcus faecalis Biofilm and Inhibits Its Growth on the Surface of Chicken Meat"

_viruses, 2023, doi:10.3390/v15051208_

Round 1
Reviewer 1 Report
Comments and Suggestions for Authors
This study describes the effect of newly isolated phages on the inhibition of biofilms and also Enterococcus in chicken meat. Overall, the phages were well characterized, but there are few doubts that are needed to be explained.
Major comments:
1. In introduction, the antibiotic resistance issue was stated and possibility of using phage as alternative. But, in this study, the antibiotic-resistant bacteria were not used. Then, The introduction should be revised according to major works conducted in this study.
2. In Figure 1F, phages were sensitive to chloroform. In general, chloroform is not effective against phages, which can be useful in the process of phage isolation and storage. Furthermore, chloroform is often used to extract and purify phages from bacterial cultures. The results should be further discussed with this issue.
3. The biofilms were inhibited regardless of phage titers. It should be also further discussed.
4. In biofilm assay, the biofilms used in this study may not be true biofilms rather just attached cells. This should be confirmed.
5. CV assay is not sensitive method to evaluate the biofilm quantitatively, specifically highly variable for loosely adhered cells. Then, this result should be supported by additional method such as plate counting results.
6. The phages were effective at low temperature. Low temperatures can slow down the bacterial growth and metabolism. This might affect the rate of phage infection and replication, eventually lytic activity. The stability of phages at 4 degree C dose not mean they are active at low temperature. This should be further discussed
Minor comments;
L 52: Huazhong Agricultural University Huazhong Agricultural University à delete repeated words
L188: Bioogical to Biological
Comments on the Quality of English Language
Minor editing is required.
Author Response
Major comments:
- In introduction, the antibiotic resistance issue was stated and possibility of using phage as alternative. But, in this study, the antibiotic-resistant bacteria were not used. Then, The introduction should be revised according to major works conducted in this study.
Thank you for your advice. We have revised the abstract, please refer to the manuscript.
- In Figure 1F, phages were sensitive to chloroform. In general, chloroform is not effective against phages, which can be useful in the process of phage isolation and storage. Furthermore, chloroform is often used to extract and purify phages from bacterial cultures. The results should be further discussed with this issue.
Line 299: The phage WH1 was sensitive to chloroform, and we speculated that the capsid might contain liposomes.
- The biofilms were inhibited regardless of phage titers. It should be also further discussed.
Line 317: Our results showed that biofilms were inhibited regardless of phage titer. Phage can reduce the number of bacteria, thus affecting the formation of biofilm, indicating that phage has the ability to inhibit the formation of biofilm. In the future, we will further investigate the mechanism of action of phages and biofilm exopolysaccharides.
- In biofilm assay, the biofilms used in this study may not be true biofilms rather just attached cells. This should be confirmed.
We determined that the biofilm used in this study is a true biofilm, a white membrane at the bottom of the 12-well plate.
- CV assay is not sensitive method to evaluate the biofilm quantitatively, specifically highly variable for loosely adhered cells. Then, this result should be supported by additional method such as plate counting results.
We added an experiment, as shown in Figure 5C. We measured the amount of bacteria in different groups. We found that the bacteria population in the phage addition group was significantly reduced.
- The phages were effective at low temperature. Low temperatures can slow down the bacterial growth and metabolism. This might affect the rate of phage infection and replication, eventually lytic activity. The stability of phages at 4 degree C dose not mean they are active at low temperature. This should be further discussed
Thank you for your advice.
Line 337: Our results demonstrated that phage WH1 can inhibit bacteria at 4 °C, and it is speculated that phage can still recognize and adsorb sensitive bacteria at 4 °C, suggesting that phage can control pathogenic bacteria in frozen meat products.
Minor comments;
L 52: Huazhong Agricultural University Huazhong Agricultural University à delete repeated words
Thank you very much for your review. I have deleted the repeated words.
L188: Bioogical to Biological
Thank you. I have revised this word.

Reviewer 2 Report
Comments and Suggestions for Authors
The manuscript reports characterization of a new siphovirus of Enterococcus faecalis which is characterized by sequence analysis, and growth properties with special attention to attacking biofilm. The phage is proposed as a candidate for use as a bactericide, such as a preservative in stored poultry meat, or for treatment of surfaces in poultry processing plants.
Phages as a bactericide are a viable option. The manuscript is somewhat muddled by copying oft repeated discussion points from phage therapy papers that are not particularly relevant. For example, no one uses antibiotics as a preservative of stored meat, so the meme about how antibiotics have disadvantages and ought to be replaced by phages is irrelevant. Similarly, lysins attack cell wall, not biofilm. So the emphasis given to the lysin doesn't make much sense. However, to be fair, even with the best annotation practices, the likelihood of being able to attribute anything about biofilm interaction to any gene sequence isn't very high at this point in time.
The manuscript contains a number of relatively standard characterization studies of the phage and the sequence, and in that respect seems reasonably thorough. A somewhat novel aspect of the manuscript is that it provides assays of the phage breaking down biofilm. Phage WH1 shows an intriguingly strong capability to do so. However, it's not obvious that "removing biofilm" from spoiled meat has any real application, since the goal is not to salvage spoiled meat, but to keep meat from getting spoiled in the first place. The more obvious assay of treating with phage, inoculating with small amounts of Enterococcus and measuring if accumulation of the bacteria is inhibited was also done. The phage appeared to do well in that assay. However, there is no comparison to other phages. There are many closely related phages, so it is unclear if this one has any special properties. That being said, with the problem of covering host range being what it is, and the phage seeming to have reasonably good host range, the phage may turn out to have some serious value.
The only data deposited in any accessible database is the raw assembled DNA contig. The annotation is relatively weakly done and there is no annotated sequence that can be acquired for reference. I ran the DNA sequence through a more standard pipeline and I found 31 genes identified, compared to 24 that they listed. I did confirm that WH1, and its many relatives, are not temperate.
Specific issues
Your large terminase gene is interrupted. What sequence exactly did you use to make your tree? Did you remove the interruption, use one end or the other, isolate the P-loop domain? How was it aligned?
Line 113: the citation given for GeneMark is not a paper about GeneMark.
Line 130 "phage" is misspelled.
Line 174: That head form is called "oblate" or "highly oblate" not "rectangular".
fig. 4. The figure doesn't show much, since the phages are all essentially the same. The track labeled WH1 doesn't have the pattern of genes in your annotation. Is it possible that track 2 is actually WH1? The criss crossing matching patterns make it difficult to interpret the figure. The criss crossing is only there because the phages weren't circularly permuted the same before making the figure.
fig. 5. I think, from your methods section, what you called the "final" phage titer is what we usually call the "initial" phage titer. I.e., it's the amount of phage added to the well before being incubated with the bacteria, not the amount of phage that grew during the incubation. In any case, the only thing that makes sense here is to indicate the amount of phage added at the beginning of the incubation.
Table 2. These should be submitted to GenBank. Preferably note a protein family matched in a public protein family database. That allows someone to look up the family, find a literature citation, and distinguish among a variety of different families having the same general description.
I encourage submitting the phage to some repository so that it's accessible to other investigators.
Author Response
Phages as a bactericide are a viable option. The manuscript is somewhat muddled by copying oft repeated discussion points from phage therapy papers that are not particularly relevant. For example, no one uses antibiotics as a preservative of stored meat, so the meme about how antibiotics have disadvantages and ought to be replaced by phages is irrelevant. Similarly, lysins attack cell wall, not biofilm. So the emphasis given to the lysin doesn't make much sense. However, to be fair, even with the best annotation practices, the likelihood of being able to attribute anything about biofilm interaction to any gene sequence isn't very high at this point in time.
Thank you for your advice. We have deleted some of the lysin statements in the discussion.
However, it's not obvious that "removing biofilm" from spoiled meat has any real application, since the goal is not to salvage spoiled meat, but to keep meat from getting spoiled in the first place.
Thank you for your review. We evaluated the bacteriostatic effect of bacteriophage WH1 at 4 ° C rather than removing biofilm from spoiled meat.
The only data deposited in any accessible database is the raw assembled DNA contig. The annotation is relatively weakly done and there is no annotated sequence that can be acquired for reference. I ran the DNA sequence through a more standard pipeline and I found 31 genes identified, compared to 24 that they listed. I did confirm that WH1, and its many relatives, are not temperate.
Thank you for your advice. I admire you for your excellent analytical skills. We have annotated the other seven genes.
Specific issues
Your large terminase gene is interrupted. What sequence exactly did you use to make your tree? Did you remove the interruption, use one end or the other, isolate the P-loop domain? How was it aligned?
The gene of the terminase large subunit (Enterococcus phage EF653P5, GenBank: OP172800.1) we referred to found that our gene was divided into two segments, which were spliced together according to the reference genes to make a phylogenetic tree again (Figure 3). Align selected block by ClustalW. We found that Enterococcus phage EF-P29 (GenBank: KY303907.1) major subunit gene was also divided into two segments, even with non-coding amino acids in the middle.
Line 113: the citation given for GeneMark is not a paper about GeneMark.
Thank you, We have revised the section.
Line 130 "phage" is misspelled.
Thank you, this seperate word has been revised.
Line 174: That head form is called "oblate" or "highly oblate" not "rectangular".
Thank you, This content has been changed to “oblate”
fig. 4. The figure doesn't show much, since the phages are all essentially the same. The track labeled WH1 doesn't have the pattern of genes in your annotation. Is it possible that track 2 is actually WH1? The criss crossing matching patterns make it difficult to interpret the figure. The criss crossing is only there because the phages weren't circularly permuted the same before making the figure.
You are right. We have revised this figure.
fig. 5. I think, from your methods section, what you called the "final" phage titer is what we usually call the "initial" phage titer. I.e., it's the amount of phage added to the well before being incubated with the bacteria, not the amount of phage that grew during the incubation. In any case, the only thing that makes sense here is to indicate the amount of phage added at the beginning of the incubation.
Thank you. I have revised the “final” to “initial”.
Table 2. These should be submitted to GenBank. Preferably note a protein family matched in a public protein family database. That allows someone to look up the family, find a literature citation, and distinguish among a variety of different families having the same general description.
Thanks for your advice, I will submit these databases to GenBank for sharing.
I encourage submitting the phage to some repository so that it's accessible to other investigators.
We will submit this phage to some repositories for other researchers to research.

Reviewer 3 Report
Comments and Suggestions for Authors
The authors after characterizing phylogenetically , proceeding to the genome sequencing and electron microscopy analyses of phage 16 vB_EfaS_WH1 (WH1) isolated from chicken feces ,they found that it as a novel phage belonging in the family Siphoviridae. Yet, they found that this phage was active and had the capacity to remove Enterococcus faecalis biofilms in chicken meat .
the paper is well written and English language is fine . However the term …conditional …pathogens( first line of Abstract ) I think that it should be better to be replaced by .. potential…pathogens.
the paper is based on a well designed protocol and a hard metrology .
the results are evaluated and discussed in depth based on a rich bibliography.
the paper should be of high interest to veterinarians , poultry science specialists and industrials
my suggestion is to ACCEPT and publish this paper
Author Response
Thank you for your review and for your recognition of my work. I have replaced the word“conditional”(first line of Abstract) with "potential".
Reviewer 4 Report
Comments and Suggestions for Authors
I have reviewed the manuscript Virulent phage vB_EfaS_WH1 removes Enterococcus faecalis 2 biofilm and inhibits its growth on the surface of chicken meat, and these are my comments:
General comment:
One of the main properties of antibacterial agents is their lack of toxicity to humans whilst being toxic to bacteria. Are there any known adverse effects of this phage (or phages in general) on humans? Please mention something on this topic in the manuscript. Please include some information on the specificity of phages in the manuscript.
Specific comments:
Lines 41-43: Does this apply to broiler chicken farms only?
Line 63: Change to “the filtrate”
Line 70: Please mention the number of bacterial species included in the host range of bacterial indicator strains.
Line 126: Remove bracket after “TEM”.
Line 130: Change to “phage”
Line 173: Change to “Phages”
Lines 177-186: Phage WH1 appears to only infect Enterococcus faecalis strains. Why is this evidence of the wide host range of this phage? Please clarify.
Line 298: Change to “phages”
Line 301: Change to “conserved”
The quality of English language in this manuscript is generally high, although a few corrections will have to be made.
Author Response
One of the main properties of antibacterial agents is their lack of toxicity to humans whilst being toxic to bacteria. Are there any known adverse effects of this phage (or phages in general) on humans? Please mention something on this topic in the manuscript.
Previous studies have shown that phages ingested by healthy individuals are safe, ac-tive, and accumulate in the lower intestine, inhibiting human IBD-associated gut mi-crobes [58]. A review of phage therapy shows that phage therapy is generally safe in 20 animal studies, 35 clinical case reports, and 14 clinical trials [59].
Please include some information on the specificity of phages in the manuscript.
Line 39: Bacteriophages are a class of viruses that attack bacteria and are highly specific…
Specific comments:
Lines 41-43: Does this apply to broiler chicken farms only?
This statement is to describe the broiler farm enterococcus faecalis resistance rate is high, there is a risk of vertical transmission, is a phenomenon. It is not said that the bacteriophage applies to broiler farms.
Line 63: Change to “the filtrate”
Thank you, this content has been revised.
Line 70: Please mention the number of bacterial species included in the host range of bacterial indicator strains.
Thank you, this content has been revised.
The host range of the purified WH1 against the 30 bacterial indicator strains included 20 strains of Enterococcus faecalis, 1 strain of Enterococcus faecium, 1 strain of Clostridium perfringens, 1 strain of Staphylococcus aureus, 2 strains of Escherichia coli, 4 strains of salmonella and 1 strain of Klebsiella…
Line 126: Remove bracket after “TEM”.
Thank you, this content has been revised.
Line 130: Change to “phage”
Thank you, this content has been revised.
Line 173: Change to “Phages”
Thank you, this content has been revised.
Lines 177-186: Phage WH1 appears to only infect Enterococcus faecalis strains. Why is this evidence of the wide host range of this phage? Please clarify.
Thank you. Phage WH1 can infect 20 enterococcus faecalis species except the host bacterium EF01, suggesting that WH1 has a multi-host range in enterococcus faecalis species in addition to infection with the host bacterium. Testing for other species of bacteria was to test whether WH1 had the ability to cross-species infection, and the result was no.
Line 298: Change to “phages”
Thank you, this content has been revised.
Line 301: Change to “conserved”
Thank you, this content has been revised.

Reviewer 5 Report
Comments and Suggestions for Authors
My compliments to the authors. The article I reviewed seemed particularly interesting to me. Great search setup, as far as I can judge. In the pdf file that I enclose I have inserted my small corrections of typos and two comments that I submit to the Authors. See the file that I enclose.

The text requires the correction of some spelling errors.
Author Response
Thank you for reviewing this manuscript and for recognizing my work.
Line 97: “The one-step growth curve protocol has been described previously [20]. Briefly, E. faecalis EF01 was cultured to the logarithmic stage (optical density at 600 nm [OD600] = 0.5).” It seems that the Authors have evaluated only the viable form of Enterococcus, certainly more sensitive to stress. Why didn't they also consider the less vital, more resistant to stress forms?
We considered that when OD600 = 0.5, E. faecalis had a better growth state and might be more sensitive to phage, which might be the reason why we chose this condition for one-step growth curve measurement.
Line 266: “The finding provided evidence that phage WH1 can effectively destroy the biofilm formed by E. faecalis EF01.” I suggest to the Authors to better explain by which ways WH1 phages prevent biofilm formation. We know that biofilms are essentially made up of EPS and that it is the bacteria themselves that produce it, within a few days. The reduction of biofilms induced by WH1 phages would therefore be induced by blocking the growth of Entrococcus and NOT by direct action of viruses on EPS, is that correct?It is better to explain this mechanism better, which is also illustrated in the next chapter.
The finding provided evidence that phage WH1 can effectively reduce the biofilm formed by E. faecalis EF01. We added an experiment, as shown in Figure 5C.
Line 317: Our results showed that biofilms were inhibited regardless of phage titer. Phage can reduce the number of bacteria, thus affecting the formation of biofilm, indicating that phage has the ability to inhibit the formation of biofilm. In the future, we will further investigate the mechanism of action of phages and biofilm exopolysaccharides.
Line 297: I don't quite understand: is the family Siphoviridae or Siphonoviridae? or are they two different families?
This should be the Siphoviridae.
Line 302: “The tail fiber of phages contains receptor binding proteins that of phage to bind with cell surface receptors.” Here it seems to me that a verb is missing from the sentence. To check.
The tail fiber of phages contains receptor binding proteins that allow phage to bind with cell surface receptors.

Round 2
Reviewer 1 Report
Comments and Suggestions for Authors
This has been well revised based on the reviewers' comments.
Comments on the Quality of English LanguageModerate editing of English language
Reviewer 2 Report
Comments and Suggestions for Authors
I have no further criticisms, other than to note that in line 175 I think they mean "plaques" where they say "phages".